# Bmal1 Regulates Prostate Growth via Cell-Cycle Modulation

**DOI:** 10.3390/ijms231911272

**Published:** 2022-09-24

**Authors:** Masakatsu Ueda, Jin Kono, Atsushi Sengiku, Yoshiyuki Nagumo, Bryan J. Mathis, Shigeki Shimba, Makoto Mark Taketo, Takashi Kobayashi, Osamu Ogawa, Hiromitsu Negoro

**Affiliations:** 1Department of Urology, Graduate School of Medicine, Kyoto University, Kyoto 606-8507, Japan; 2Department of Urology, Shizuoka General Hospital, Shizuoka 420-8527, Japan; 3Sengiku Urology Clinic, Moriyama 524-0045, Japan; 4Department of Urology, Faculty of Medicine, University of Tsukuba, Tskuba 305-8575, Japan; 5International Medical Center, University of Tsukuba Affiliated Hospital, Tsukuba 305-8576, Japan; 6Department of Health Science, School of Pharmacy, Nihon University, Chiba 245-8555, Japan; 7iACT-Colon Cancer Project, Kyoto University Hospital, Graduate School of Medicine, Kyoto University, Kyoto 606-8501, Japan; 8Tazuke Kofukai, Medical Research Institute, Kitano Hospital, Osaka 530-8480, Japan; 9Department of Urology, Japanese Red Cross Otsu Hospital, Otsu 520-0046, Japan

**Keywords:** circadian, clock, *Cdkn1a*, p21, development

## Abstract

The circadian clock system exists in most organs and regulates diverse physiological processes, including growth. Here, we used a prostate-specific Bmal1-knockout mouse model (pBmal1 KO: *PbsnCre+*; *Bmal1^fx/fx^*) and immortalized human prostate cells (RWPE-1 and WPMY-1) to elucidate the role of the peripheral prostate clock on prostate growth. Bmal1 KO resulted in significantly decreased ventral and dorsolateral lobes with less Ki-67-positive epithelial cells than the controls. Next, the cap analysis of gene expression revealed that genes associated with cell cycles were differentially expressed in the pBmal1 KO prostate. *Cdkn1a* (coding p21) was diurnally expressed in the control mouse prostate, a rhythm which was disturbed in pBmal1 KO. Meanwhile, the knockdown of BMAL1 in epithelial RWPE-1 and stromal WPMY-1 cell lines decreased proliferation. Furthermore, RWPE-1 BMAL1 knockdown increased G0/G1-phase cell numbers but reduced S-phase numbers. These findings indicate that core clock gene Bmal1 is involved in prostate growth via the modulation of the cell cycle and provide a rationale for further research to link the pathogenesis of benign prostatic hyperplasia or cancer with the circadian clock.

## 1. Introduction

The circadian clock generates the cyclical day–night somatic rhythm and modulates diverse physiological processes, including organ growth. The central clock is localized in the suprachiasmatic nucleus (SCN), from where it controls peripheral clocks located in organs, tissues, and even cells [1]. The circadian master rhythm is generated by transcription–translation feedback loops, consisting of a conserved family of clock genes, with Bmal1 playing a particularly critical role in reproductive endocrinology [2].

The prostate is a male reproductive gland responsible for 30–35% of semen composition with a pass through for the urethra. Prostate weights are usually less than 20 g in adults, while the exact size varies by individual [3]. Mid-to-late life mechanisms of enlargement mainly occur through hormones such as testosterone and estrogen [4] but also through inflammation [5], oxidative stress [6] and insulin resistance [7]. However, any associations between clock genes and prostatic growth via the modulation of these causative processes are largely unknown. To investigate such a role for peripheral clock genes in prostatic hyperplasia, the isolation of mechanistic pathways through targeted deletions has become a standard process for molecular studies. For example, the global Bmal1 knockout mouse is characterized by decreased testosterone production in Leydig cells associated with increased luteinizing hormone [8,9]. However, even with these methods, the direct influence of clock genes on hyperplasia or other types of proliferative growth is still obscure in the prostate [8,10].

As clock genes are known to link with cell cycles and modulate cellular proliferation, their expression follows an oscillatory pattern in the mouse prostate that can be exploited to elucidate the growth effect of these key regulatory genes [11]. Here, we generated prostate-specific Bmal1 knockout (pBmal1 KO) mice to detail the role of the circadian clock in prostatic growth through analyzing the phenotypes of the mice.

## 2. Results

### 2.1. Prostate-Specific Deletion of Bmal1

*PbsnCre+*; *Bmal1^fx/fx^* (prostate-specific Bmal1 knockout; pBmal1 KO) mice were generated by crossing *PbsnCre+* male mice and floxed Bmal1 (*Bmal1^fx/fx^*) female mice [12,13,14]. To validate the conditional knockout efficacy of *Bmal1*, real-time quantitative RT-PCR was performed. The *Bmal1* expression levels in all three prostate lobes of pBmal1 KO mice were significantly lower than those of control *Bmal1^fx/fx^* mice (Figure 1A), while the expression levels of *Bmal1* in other organs, such as the bladder and the kidney, were maintained. The knockout of BMAL1 was also confirmed by immunoblotting (Figure 1B).

### 2.2. Evaluation of Prostatic Weight in pBmal1 KO Mice

Prostatic weights were compared using the prostatic index (prostatic weight/body weight, mg/g) in 20-week-old mice. The prostatic indices of pBmal1 KO mice were slightly smaller than control *Bmal1^fx/fx^* mice for ventral prostates (VPs) and dorsolateral prostates (DLPs), while the indices for anterior prostates (APs) did not significantly differ (Figure 1C). The prostates of pBmal1 KO mice were also assessed histopathologically, showing no obvious changes, such as cell death or atrophy, compared with the prostates of control mice (Appendix A). Serum testosterone levels did not significantly differ between groups (Appendix A).

### 2.3. Immunostaining of pBmal1 KO Mouse Prostates with Ki-67 Antibody

Next, the Ki-67 immunostaining of mouse prostates was performed to investigate the proliferative ability (Figure 2A). Ki-67-positive epithelial and stromal cells were counted, revealing that significantly less Ki-67-positive epithelial cells were detected in the VPs and DPs of pBmal1 KO mice compared with control *Bmal1^fx/fx^* mice (Figure 2B). This finding indicated that epithelial cells of pBmal1 KO mice were less proliferative than those of the controls.

### 2.4. Comprehensive Analysis of Differently Expressed Genes in pBmal1 KO Mouse Prostates

To reveal the mechanisms of how Bmal1 knockout produces smaller prostate sizes and weights, a cap analysis of gene expression (CAGE) was performed. The gene expression of the DLPs of 20-week-old mice was comprehensively compared between pBmal1 KO mice and control *Bmal1^fx/fx^* mice. As expected, the knockout of *Bmal1* affected the expression of major clock genes as follows: *Npas2*, *Cry1* and *Rorc* were upregulated. The differentially expressed gene (DEG) analysis identified 42 upregulated genes and 24 downregulated genes induced by the knockout (Table 1). Furthermore, the enrichment analysis using Metascape demonstrated that cell-cycle-related genes were significantly altered (Figure 3), hinting at a potential relationship between Bmal1 and cell cycles in the prostate. Additionally, *Cdkn1a*, a gene encoding p21, was upregulated in pBmal1 KO mice.

### 2.5. Oscillation of Cdkn1a and Clock Gene Expression in Mouse Prostates

To investigate the diurnal expression of *Cdkn1a* in the prostate and the association of prostate-localized Bmal1, time-course experiments over 24 h under light–dark cycles (lights on, ZT = 0, and off, ZT = 14) were performed in control *Bmal1^fx/fx^* mice and pBmal1 KO mice. The expression of *Bmal1* in control mice showed a diurnal oscillation (cosinor analysis, *p* = 0.002), whereas pBmal1 KO mice experienced decreased expression and lost oscillation (Figure 4A). The expression of *Cdkn1a*, which encodes p21, oscillated in control mice (cosinor analysis, *p* = 0.009) while expression similarly increased with disrupted oscillation in pBmal1 KO mice (Figure 4B). These findings indicate that the expression of *Cdkn1a* had a diurnal rhythm in the prostate and that its expression was negatively modulated by *Bmal1*.

### 2.6. Expression of p21 in pBmal1 KO Mice

Next, to confirm the upregulation of the p21 protein in pBmal1 KO mouse prostates, the immunoblotting and immunohistochemistry of p21 were conducted. An increased expression of p21 was observed in pBmal1 KO (Figure 5A), and the positive staining of stromal cells was only observed in pBmal1KO mouse prostates (Figure 5B), suggesting that the knockout of BMAL1 negatively affected the cell cycle and led to growth inhibition of the prostate.

### 2.7. Proliferation Assay (WST-8 Assay) in BMAL1-Knockdown Immortalized Human Prostate Cells

To further investigate the role of BMAL1 in the growth of the prostate, a WST-8 assay was performed using immortalized human prostate cells. Here, BMAL1 knockdown (Appendix A) reduced cell growth in both epithelial and stromal cells (Figure 6A,B), indicating that BMAL1 was required for optimal proliferation at the cellular level even in transformed cells.

### 2.8. Cell-Cycle Analysis via Flow Cytometry in BMAL1-Knockdown Immortalized Human Prostate Cells

A flow cytometry analysis was performed to elucidate the cell-cycle phase distributions of immortalized human prostate cells. Here, the number of BMAL1-knockdown stromal cells in the G0/G1 phase was found to be increased, but that in the S phase was reduced (Figure 7A,B), in line with observed increases in p21 expression and reduced cell growth. These findings indicate that Bmal1 affected prostatic growth by modulating the cell cycle.

## 3. Discussion

The purpose of this study was to investigate the role of the peripheral clock in prostatic growth/hyperplasia using pBmal1 KO mice. In the current study, pBmal1 KO mice exhibited a mild decrease in prostatic weight and prostatic weight/body weight ratio compared with control mice. This decrease in growth in pBmal1 KO mice was also confirmed via Ki-67 immunostaining, demonstrating that Bmal1 was associated with cellular proliferation through the regulation of the cell-cycle phases. The CAGE revealed that genes associated with the cell cycle were, with regard to rhythm, differentially expressed in pBmal1 KO prostate, while *Cdkn1a* (encoding p21) was diurnally expressed in the control mouse prostate. Associations between Bmal1 and cellular proliferation were confirmed using the in vitro BMAL1-knockdown immortalized human prostate cell line, which showed reduced cellular proliferation and increased p21 expression, reflected by increased G0/G1 and reduced S phases. Collectively, the present study indicates that peripheral clock protein Bmal1 is involved in prostatic growth through the modulation of cellular proliferation via the cell cycle.

One important finding of this study is that the prostate-specific knockout of Bmal1 affected prostate growth via cell-cycle modulation. Under this condition, we found a robust p21 oscillation in *Bmal1^fx/fx^* mouse prostates, indicating that p21 expression was tightly linked with the circadian rhythm at both the organ and tissue levels [15]. On the other hand, the expression of p21 in pBmal1 KO mice was upregulated almost completely throughout the day and lost its rhythmicity. The cell-cycle analysis of immortalized human prostate cells further revealed that knocking down BMAL1 resulted in an increased proportion of G0/G1-phase cells and less S-phase cells. Since p21 is a key regulatory molecule that prevents the G1/S transition by binding to Cdk4/Cyclin D complex to modulate the cell cycle [16], the upregulation of p21 in pBmal1KO mice was in line with the growth inhibition of the prostate. Mammalian p21 has two ROR response elements (ROREs) in its promoter, themselves being regulated by circadian clock components, namely, RORa/RORc (acting as activators) and REB-ERBα/REV-ERBβ (acting as inhibitors) [15]. Our CAGE and qPCR analysis revealed the upregulation of Rorc in the pBmal1 KO mice throughout the day in addition to p21 (Appendix A). The putative link between Bmal1 and cell growth in this study is that the increased expression of Rorc via the dysregulation of the circadian clock system under pBmal1 KO induced p21 upregulation, followed by an increase in G0/G1-phase proportion and decreased S-phase time that inhibited prostatic growth. Shidaifat et al. showed that gossypol, a male antifertility agent, reduced the proliferation of human benign prostatic hyperplastic cells through G0/G1 arrest [17]; our observed association between clock genes and cell cycles in the prostate is in line with these findings.

Since Bmal1 signaling is associated with the growth of the prostate, it is possible that it may also be involved in benign prostatic hyperplasia (BPH), as disruptions in turnover rates drive organ growth, and the prostate has a much higher turnover rate compared with the seminal vesicles [18]. BPH is the most common urological disease in men over 50 years of age and affects bladder voiding [4]. Although associations between the circadian clock and the pathogenesis of prostate cancer are elucidated in the basic and clinical literature [19,20], the relationships among Bmal1 signaling, BPH and prostate cancer remain for future research to elucidate.

This study had several limitations. Firstly, the functional roles of Bmal1 in the prostate are still unclear, since male pBmal1 KO mice were as fertile as controls. This makes a detailed semen analysis a necessary component of future studies. Second, the knockout efficiency of the experiment in vitro was only examined at the mRNA level, albeit with significant decreases being observed (Appendix A). Lastly, knockdown efficiency might have been suboptimal for this model (especially with regard to the AP), since the AP region had only half the observed decrease in mRNA levels and no significant differences in size, while probasin was observed in all lobes [21].

In conclusion, prostatic *Bmal1*, a core clock gene, was involved in the growth of the prostate via the modulation of the cell cycle. The findings of this study can provide a basis for further research to elucidate the reproductive physiology of the prostate and the pathogenesis of benign prostatic hyperplasia or cancer in terms of the circadian clock.

## 4. Materials and Methods

### 4.1. Animals

*PbsnCre+*; *Bmal1**^fx/fx^* mice and *Bmal1**^fx/fx^* mice were housed under specific-pathogen-free, controlled conditions (constant room temperature and lights on 7:00 am to 9:00 pm). Food and water were given ad libitum. All animal experiments were approved by Kyoto University Institutional Animal Care and Use Committee (IACUC; permit number: MedKyo 18242) and conducted according to the guidelines for animal experimentation of the experimental animal center of Kyoto University.

### 4.2. Dissection and Weight Measurement of Mouse Prostate Glands

Experimental 20-week-old male mice were anesthetized with isoflurane and euthanized via cervical dislocation, followed by immediate en bloc dissection of the prostate, urethra, bladder, seminal vesicles, ampullary glands and proximal vas deferens. The prostate was divided into 4 lobes (VP, DP, LP and AP) under the microscope. Each lobe was rinsed and weighed immediately after removal using an electronic scale with DP and LP clumped together as DLP.

### 4.3. Histology and Immunohistochemistry of Mouse Prostate Glands

Prostate glands were obtained from 20-week-old mice. The prostates were fixed in 10% neutral buffered formalin and embedded in paraffin blocks. Serial sections (5 μm) were cut and submitted to routine hematoxylin and eosin (H&E) staining. Immunostaining with Ki-67antibody (12202; Cell Signaling Technology, Danvers, MA, USA) and p21 antibody (187; Santa Cruz Biotechnology, Dallas, TX, USA) was performed at Center for Anatomical, Pathological and Forensic Medical Researches at Kyoto University. Ki-67-positive epithelial cells were counted in each lobe (VP, LP and DP).

### 4.4. Cap Analysis of Gene Expression (CAGE)

CAGE library preparation, sequencing, mapping and gene expression analysis were performed by DNAFORM (Kanagawa, Japan). Total RNA was extracted using RNeasy Mini Kit (Qiagen, Hilden, Germany) and RNA quality was assessed using Bioanalyzer (Agilent Technologies, Santa Clara, CA, USA) before standardization to an RNA integrity number (RIN) > 7.0. RNA purity was analyzed using Nano Drop and considered good-quality when the A260/280 and A260/230 ratios were >1.7. First-strand cDNAs were transcribed to the 5′ end of capped RNA and attached to CAGE “bar code” tags, and the sequenced CAGE tags were mapped to the mouse mm9 genomes using BWA software (v0.5.9, Wellcome Trust Sanger Institute, Cambridge, UK) after discarding ribosomal or non-A-/C-/G-/T-base-containing RNAs. For tag clustering, CAGE-tag 5’ coordinates were input for Reclu clustering [22], with a maximum irreproducible discovery rate of (IDR) 0.1 and a minimum tags per million (TPM) value of 0.1. Differentially expressed gene (DEG) analyses were performed using the edgeR package in Reclu. The statistical threshold for DEGs was defined as a false discovery rate (FDR) <0.05 and absolute log2 (fold-change) >0.5. The enrichment in the DEGs was analyzed using Metascape with default settings [23].

### 4.5. Real-Time Quantitative RT-PCR Analysis

Total RNA from mouse prostate glands was extracted similarly to the CAGE samples. Complementary DNA was synthesized from 1 μg of RNA using ReverTra Ace qPCR RT Kit (TOYOBO, Osaka, Japan). Real-time quantitative RT-PCR was performed with SYBR Green PCR Master Mix (Life Technologies, Carlsbad, CA, USA) and a 7300 Real-time PCR system (Life Technologies). The thermal cycling conditions were set at 94 °C for 15 s, 60 °C for 15 s and 72 °C for 1 m. Values were adjusted relative to the expression levels of the housekeeping gene human Gapdh or mouse 18 s ribosome. The primers used are shown in Appendix A. The ΔΔCt method was adapted to evaluate the relative gene expression of the target genes.

### 4.6. Immunoblotting

Mouse prostate cell lysates were prepared with radioimmunoprecipitation assay (RIPA) buffer containing proteinase inhibitors. Protein samples (30 μg) were separated on SDS polyacrylamide gels and transferred to polyvinylidene difluoride membranes (Millipore, Bedford, MA, USA) with a Mini Trans-Blot Cell system (Bio-Rad Laboratories, Hercules, CA, USA). Membranes were blocked with 5% bovine serum albumin-diluted TBST (BSA/TBST) for 1 h and incubated with primary antibodies diluted in 1% BSA/TBST overnight, followed by incubation with the corresponding secondary antibodies diluted in 1% BSA/TBST for 50 min. Immunoreactive protein bands were visualized using enhanced chemiluminescence (SuperSignal West Pico Chemiluminescent Substrate, Thermo) and an LAS-4000 imaging system (Fujifilm Life Science, Tokyo, Japan). The following primary antibodies were used: anti-BMAL1 (Santa Cruz Biotechnology; H-170; 1:200), anti-p21 (Santa Cruz Biotechnology; 187; 1:100) and anti-beta actin (ab6276; Abcam; 1:5000). The levels of Bmal1, p21 and beta-actin (loading control) protein expression were quantified using ImageJ software (1.52u, National Institute of Health, Rockville Pike, MD, USA; http://rsb.info.nih.gov/ij/, accessed on 15 Septemer 2022). Values were first normalized to the respective loading control and were expressed relative to *Bmal1^fx/fx^* levels.

### 4.7. Cell Culture

Human prostate epithelial RWPE-1 and stromal WPMY-1 cell lines were purchased from American Type Culture Collection (Rockville, MD, USA). RWPE-1 cells were routinely cultured in keratinocyte serum-free medium (Invitrogen, Carlsbad, CA, USA), and WPMY-1 cells were cultured in Dulbecco’s modified Eagle medium (Invitrogen).

### 4.8. Cell Transduction

Cell lines were stably transduced with shRNA lentivirus and selected in the presence of 1.5 μg/mL puromycin. Lentivirus-based plasmid containing oLKO.1 shRNA sets to human Bmal1 (SHCLG-NM_001178; TRCN0000331012 (sh12), TRCN0000331014 (sh14), TRCN0000331079 (sh79)) were purchased from Sigma-Aldrich (St. Louis, MO, USA). Non-silencing shRNA (Sigma-Aldrich) was used as a negative control. The experimental procedure for shRNA transfection was performed according to the manufacturer’s protocol.

### 4.9. Cellular Proliferation Assay

Cells were seeded into 96-well plates at the density of 5000 cells per well in each medium. Proliferative activity was measured using the WST-8 assay (Dojindo, Kumamoto, Japan) using a microtiter plate reader at 450 nm.

### 4.10. Flow Cytometry

Cells were harvested, washed with cold PBS and fixed with 70% ethanol. The fixed cells were incubated with 500 μL of propidium iodide (PI; 1 mg/mL), followed by cell-cycle analysis using BD FACSAria III (Becton, Dickinson and Company, Franklin Lakes, NJ, USA).

### 4.11. Statistical Analysis

Data are shown as the means ± SE. The results were analyzed using the unpaired t-test and the one-way analysis of variance (ANOVA) with Tukey’s post hoc test when appropriate. EZR (Saitama Medical Center, Jichi Medical University), a graphical user interface for R (R Foundation for Statical Computing), and a modified version of R Commander designed to add statistical functions frequently used in biostatistics were used for all analyses [24]. *p*-values less than 0.05 were considered as significant.

## Figures and Tables

**Figure 1 ijms-23-11272-f001:**
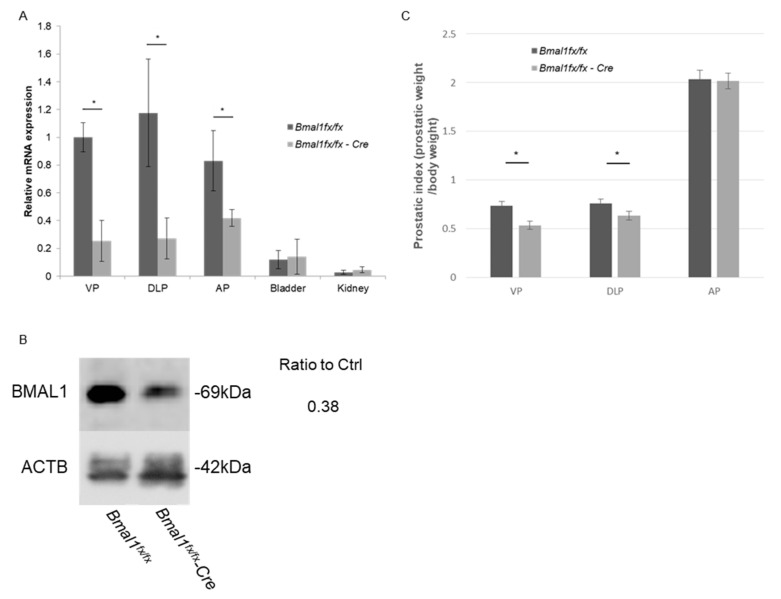
Validation of the *Bmal1* knockout in three prostatic lobes of *PbsnCre+*; *Bmal1^fx/fx^* mice via real-time PCR (**A**) and immunoblotting (**B**). (**C**) Prostatic weight comparison between 20-week-old *PbsnCre+*; *Bmal1^fx/fx^* mice and *Bmal1^fx/fx^* mice; *N* = 16 each. * *p* < 0.05 using Student’s *t*-test. VPs, ventral prostates; DLP, dorsolateral prostate; AP, anterior prostate.

**Figure 2 ijms-23-11272-f002:**
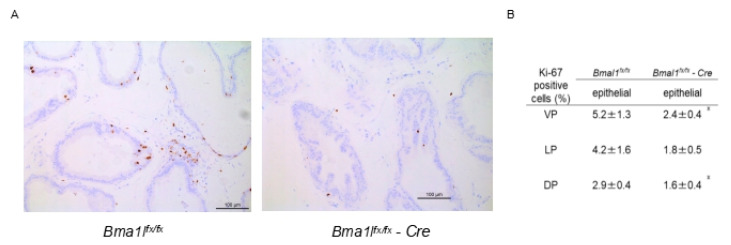
Ki-67 immunostaining of mouse prostates. (**A**) Representative image of 20-week-old mouse prostate (VP). (**B**) Percentage of Ki-67-positive cells; *N* = 6 each. * *p* < 0.05 using Student’s *t*-test.

**Figure 3 ijms-23-11272-f003:**
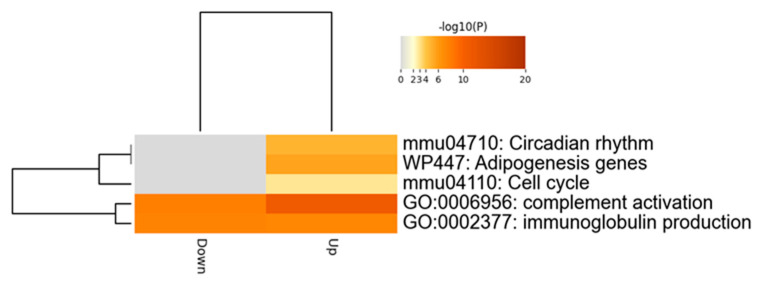
Enrichment analysis using Metascape.

**Figure 4 ijms-23-11272-f004:**
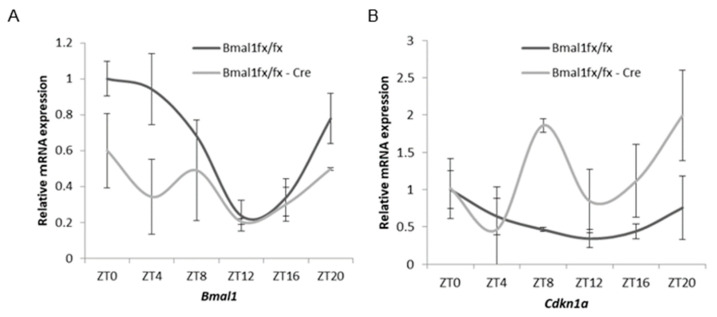
Oscillation of clock gene and *Cdkn1a* expression in mouse prostates detailing the temporal mRNA accumulation of *Bmal1* and *Cdkn1a* in the dorsolateral mouse prostate under light–dark conditions (*N* = 3 for each time point). (**A**) *Bmal1*. (**B**) *Cdkn1a*. *p*-values with cosinor analysis in *Bmal1* and *Cdkn1a* were 0.002 and 0.009 in *Bmal1^lx/lx^* mice and 0.296 and 0.026 in pBmal1 KO mice, respectively.

**Figure 5 ijms-23-11272-f005:**
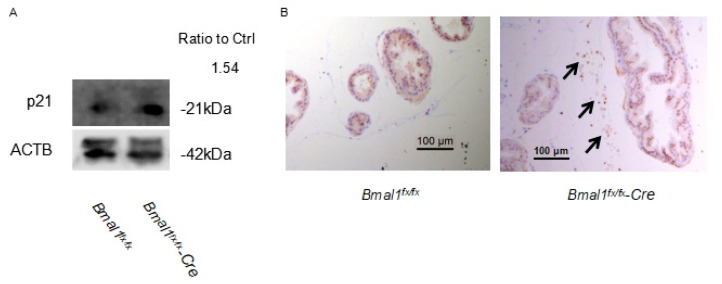
Representative images of protein expression levels of p21 in the prostate. (**A**) Immunoblotting of dorsolateral prostates at ZT16. (**B**) Immunohistochemistry of ventral prostates. Stromal cells showed positive staining in pBmal1 KO mice (shown by arrows) but little in *Bmal1^fx/fx^* mice.

**Figure 6 ijms-23-11272-f006:**
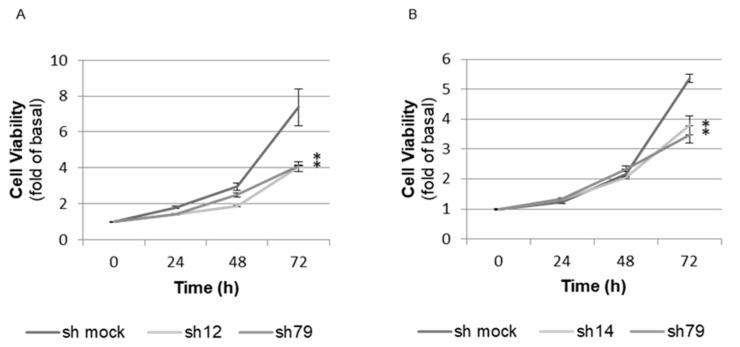
Proliferation assay (WST-8 assay) in *BMAL1*-knockdown immortalized human prostate cells. (**A**) Prostate epithelial cells, RWPE-1. (**B**) Prostate stromal cells, WPMY-1. Both BMAL1-knockdown cells showed more delayed proliferation than mock cells. * significantly decreased with respect to the control. *p* < 0.05 using one-way ANOVA with Tukey’s post hoc test (n = 3).

**Figure 7 ijms-23-11272-f007:**
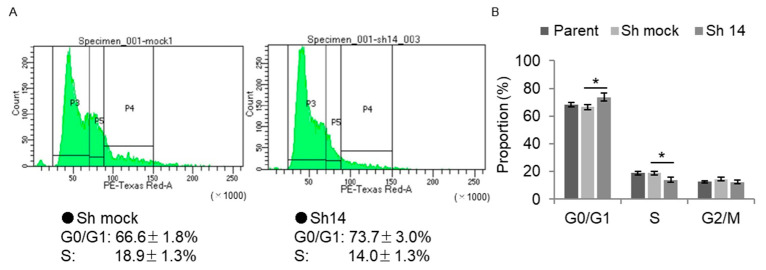
Cell-cycle analysis via flow cytometry in *BMAL1*-knockdown immortalized human prostate cells. (**A**) Representative graphics of control cells and *BMAL1*-knockdown cells. The cell cycle was determined with propidium iodide (PI) staining. (**B**) Calculated proportion of cells in G0/G1, S and G2/M phases (*N* = 6). * *p* < 0.01 using one-way ANOVA with Tukey’s post hoc test.

**Table 1 ijms-23-11272-t001:** A list of differentially expressed genes between *PbsnCre+*; *Bmal1^fx/fx^* mice and *Bmal1^fx/fx^* mice, as discovered using CAGE. Dorsolateral prostates of 20-week-old mice were analyzed (*N* = 3 each).

Up-Regulation				Down-Regulation	
Gm16971	Ighv10-3	Cyp2b10	Cfd	Mpz	Igkv8-18
Igkv1-135	Gm16698	Gm14017	Cdkn1a	Fam195b	Gm16710
Gm13253	Npas2	Gm16948	Cry1	Gbp8	Gm16700
Gm11755	Abpb	Iglv2	Rorc	AC125484.1	Gm16829
Gm16708	Igkv15-103	Igkv12-89	Iglv1	Gm16717	Igkv4-61
Gm16792	Iglc2	Thrsp	Gm14326	Cyp2e1	Rpl9-ps4
Ccne2	Ephx2	Igkv16-104		Ighv1-62-2	Igkv4-78
Retn	Gm16842	Ifi27l2a		Igkv8-27	Gm16949
SNORA19	Plin1	Fabp4		Igkv5-39	Hp
Gm5417	Car3	Pttg1		Igkv1-110	Abpz
Igkv1-117	Adipoq	Prss28		Zfp959	Psca
Igkv6-13	Gm4167	Gm6644		Gm10243	Gm6793

## Data Availability

Not applicable.

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
