# Peer review of "Bmal1 Regulates Prostate Growth via Cell-Cycle Modulation"

_ijms, 2022, doi:10.3390/ijms231911272_

Round 1
Reviewer 1 Report
The paper entitled Bmal1 Regulates Prostate Growth via Cell Cycle Modulation is presented for peer review. Circadian rhythm is found in most of living species worldwide. Authors surely noted that The circadian clock generates the cyclical day-night somatic rhythm and modulates diverse physiological processes, including organ growth. Prostate is known to be one of mitotically active organs of human. By the way, associations between clock genes and prostatic growth via modulation of these causative processes are largely unknown. Authors generated prostate-specific Bmal1 knockout (pBmal1 KO) mice to detail the role of the circadian clock in prostatic growth through analyzing the phenotypes of the mice.
Paper is interesting to read. I have several suggestions.
1. Have you verified knockout by western also or qPCR alone. If so, have you compared mBmal1 protein levels in prostate and other organs?
2. Have you used other proliferative markers like PCNA to prove the decrease in KO cells?
3. Please indicate light conditions for mBmal1 expression analysis (lines 109-117). Light regimen could affect Bmal1 expression time course.
4. Please verify p21 expression in protein level or explain in the text.
5. To prove interaction of p21 and Bmal1 I recommend to use ChiP method or ChiP seq.
Reviewer 2 Report
Comments: Manuscript is very smart work. Few issues should to be studied.
1- Nuclear regulation was not studied. for example, cyclin D should have been revised by western blot
2- Histological sections for both control and pBmal1 KO mice stained by H*E should have been provided to understand histologically if there were atrophy or cell death.
3- In Table 1. “Up and Dawn” should to be changed into "up regulation" and "Down regulation"
4- Table caption should be written up to table, not below table.
5- The mechanism by which BMAL1 can affect growth of prostate was not clearly explained.
6- In Figure 5. graph A and B " X/Y lines should be observed.
7- The result of p21 gene expression was not provided in text.
8- There is no explanation was provided for why BMAL1 reduced S phase and increased G0/G1 phase.
9- The primers used in studied genes were missed. primers should have been summarized in table
Round 2
Reviewer 2 Report
Manuscript was revised point by point according to reviewer comments and it is more acceptable NOW.